# COMPUTATIONAL REASONING OF LARGE LANGUAGE MODELS

## ABSTRACT

With the rapid development and widespread application of Large Language Models (LLMs), multidimensional evaluation has become increasingly critical. However, current evaluations are often domain-specific and overly complex, limiting their effectiveness as cross-domain proxies for core capabilities. To address these limitations and enable a unified and simple evaluation framework, an ideal proxy task should target a basic capability that generalizes across tasks and is independent of domain-specific knowledge. Turing machine provides a powerful theoretical lens by reducing complex processes to basic, domain-agnostic computational operations. This perspective offers a principled framework for evaluating foundational computational abilities essential to a wide range of tasks, particularly those involving complex, multi-step reasoning such as mathematics. Motivated by this abstraction, we introduce **Turing Machine Bench**, a benchmark designed to assess the ability of LLMs to **strictly follow rules** and **accurately manage internal states** for multi-step, referred to as **computational reasoning**. TMBench incorporates four key features: self-contained and knowledge-agnostic reasoning, a minimalistic multi-step structure, controllable difficulty, and a solid theoretical foundation based on Turing machine. Empirical results demonstrate that TMBench serves as an effective proxy for evaluating computational reasoning on representative LLMs. It produces clear step-wise accuracy curves, revealing LLMs' ability to execute multi-step reasoning processes. By analyzing performance trends across TMBench and established reasoning benchmarks, we find strong correlations with real-world tasks, bridging real-task evaluation with basic ability assessment. These findings suggest that TMBench holds potential as a cross-domain dimension for evaluating reasoning in LLMs. Code and data are available at Repo.

## 1 INTRODUCTION

Recent progress in pre-training, post-training, and scaling has significantly advanced the capabilities of LLMs. These models now demonstrate remarkable capabilities not only in traditional natural language processing tasks, such as text classification and machine translation Devlin et al. (2019), but also in more complex cognitive domains, including advanced reasoning Rein et al. (2024); Huang et al. (2024), code generation Jimenez et al. (2023); Chen et al. (2021a), instruction following La Malfa et al. (2024), multimodal understanding Wu et al. (2023), and even scientific discovery Boiko et al. (2023). While these advancements significantly broaden the scope of potential applications, the growing complexity and diversity of cognitive tasks undertaken by LLMs present substantial challenges in accurately and reliably evaluating their true capabilities. The intelligence of LLMs can be mainly evaluated along two key dimensions: (1) knowledge and comprehension, and (2) reasoning and decision-making. First, knowledge and comprehension benchmarks primarily focus on the ability of LLM to apply its internal knowledge to understand and solve user questions. These skills are typically evaluated through large-scale question answering tasks (e.g., MMLU Hendrycks et al. (2020), TriviaQA Joshi et al. (2017)). Second, reasoning is the process of inferring and justifying conclusions from a set of premises Arkoudas (2023). In particular, reasoning encompasses a broad range of tasks, including logical deduction (e.g., Big-Bench Hard bench authors (2023)), mathematical reasoning (e.g., AIME MAA (2024)), formal proof (e.g., miniF2F Zheng et al. (2021), ProofNet Azerbayev et al. (2023)), programming (e.g., HumanEvalChen et al. (2021a)) and computational reasoning(e.g., Arithmetic Lai et al. (2024), CodeSimulation La Malfa et al. (2024)). Specifically, computational

reasoning refers to the ability to accurately interpret formal rules and execute multi-step computational operations without external tools. This ability is fundamental to modern sciences, which are generally founded on rule-based systems. Disciplines such as mathematics, physics, and chemistry rely on axioms, theorems, and laws, collectively referred to as rules. At the most fundamental level, arithmetic represents the simplest form of such rules. Although LLMs are not inherently proficient at direct numerical computation, they have demonstrated the ability to perform accurate arithmetic by simulating a Turing machine designed for arithmetic Lai et al. (2024). This suggests that LLMs have the potential to faithfully execute formal rules in multi-step reasoning tasks. By assessing the results across these two dimensions, we obtain a more comprehensive understanding of an LLM's intelligence. However, these dimensions often overlap in practice. For instance, solving a mathematical problem may still rely on background knowledge encoded during pre-training, blurring the distinction between memorization (knowledge) and computation (reasoning).

To evaluate the computational reasoning ability of LLMs, we turn to the foundational paradigm of theoretical computer science: the Turing machine. Its minimalist design and computational universality provide a precise framework to measure a model's ability to follow rules and deterministic state transitions. We introduce **TMBench**, a benchmark for evaluating computational reasoning of LLMs by simulating the operation of an $m$-Tag Turing machine. The $m$-Tag machine consists of production rules and a dynamic queue. At each step, it reads the queue head, appends the corresponding production symbols to the tail, and deletes the first $m$ characters. Iterating this process and comparing the model's execution trace with ground truth yields a quantitative measure of reasoning ability as an executor. TMBench has four key features: (1) **Self-contained reasoning**: all tasks are solvable from first principles without external knowledge; (2) **Minimalistic multi-step structure**: each step is necessary, interpretable, and yields a verifiable result; (3) **Controllable difficulty**: rule sets, $m$ values, step counts, and input lengths can be tuned to vary complexity; (4) **Computational generality**: any computable function can be represented by a Turing machine, making TMBench a principled proxy for rule-following and state management. Our main contributions are summarized as follows:

1. We propose the TMBench to evaluate the computation reasoning ability of LLMs. Evaluations are conducted on a broad range of recent open-source models, from 0.6B to 671B parameters, as well as proprietary models such as Gemini, GPT, Grok, and Claude.

2. Empirical results demonstrate that TMBench strongly correlates with established reasoning benchmarks and real-world task performance, highlighting its practical relevance. It effectively captures LLMs' multi-step reasoning abilities through clear step-wise accuracy trends, making it a promising cross-domain metric for evaluating computational reasoning.

3. To investigate TMBench's characteristics, we conduct ablation studies on unbounded-step execution, decoding temperature, alphabet type, task difficulty and SFT. Results show that LLM inevitably fails with increasing steps due to its autoregressive nature. Its stable performance across different alphabets suggests reliance on reasoning rather than statistics. Varying deletion counts enables fine-grained control of task difficulty, demonstrating the benchmark's scalability.

## 2 RELATED WORK

### 2.1 LLM BENCHMARKS

Benchmarks for large language models can be primarily categorized into two types: closed-ended benchmarks, which provide definitive answers, and open-ended benchmarks, which rely on human preference. For closed-ended benchmarks, a wide range of topics is addressed, including language understanding, mathematics, coding, reasoning, hallucination, toxicity, and stereotypes. Notable benchmarks in this category include MMLU Hendrycks et al. (2020), HellaSwag Zellers et al. (2019b), GSM-8K Cobbe et al. (2021a), HELM Liang et al. (2022), BigBench bench authors (2023), AGIEval Zhong et al. (2023), HumanEval Chen et al. (2021b), and ToxicChat Lin et al. (2023). Beyond closed-ended questions, benchmarks also include open-ended questions that with human preference. These questions are typically assessed through expert ratings or crowd-sourced evaluations. The recent trend includes utilizing GPT-4 for approximating human judgment Chiang & Lee (2023), with notable instances being MT-Bench Zheng et al. (2023) and AlpacaEval Li et al.

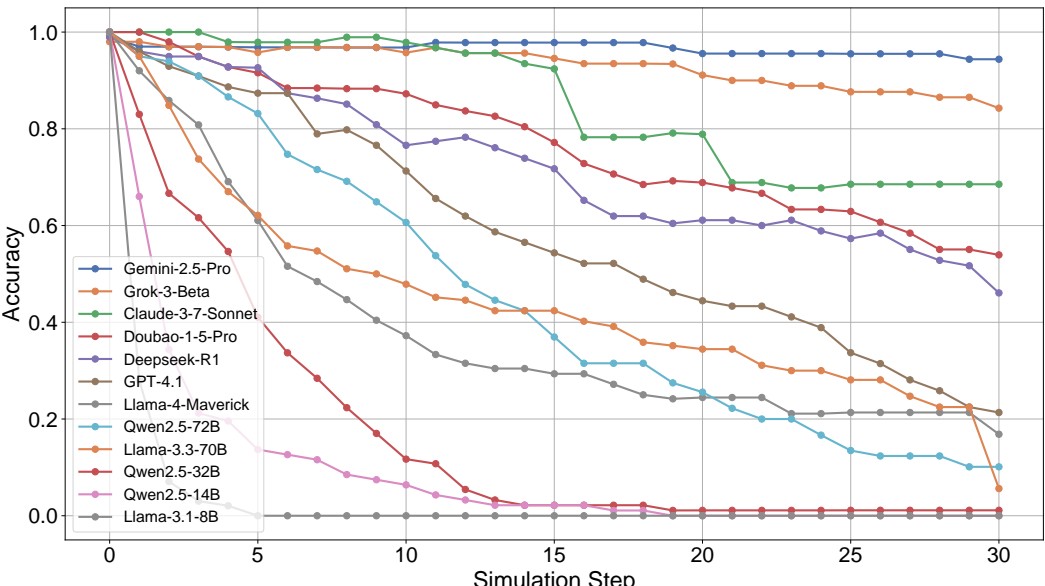

Figure 1: Illustration of the multi-step performance curve on TMBench across a diverse range of both open-source and proprietary LLMs. Proprietary models demonstrate advantages in computational reasoning abilities, but accuracy still decreases as steps increase.

(2023). In addition to static benchmarks, dynamic benchmarks featuring continuously updated questions are also available. These questions may be drawn from annual examinations, weekly online contests such as Codeforces Li et al. (2022); Huang et al. (2023), or collaborative platforms like ChatbotArena Zheng et al. (2023). Some studies have explored leveraging live human interaction for reinforcement learning from human preferences Bai et al. (2022); Ouyang et al. (2022); Touvron et al. (2023). In this paper, we introduce Turing Machine Bench, a self-contained, multi-step benchmark for computation reasoning that avoids the influence of external knowledge.

## 2.2 MULTI-STEP REASONING

Recent large language models demonstrate remarkable performance in solving complex reasoning tasks Lewkowycz et al. (2022). Multi-step reasoning, essential for tackling such tasks, has gained significant attention, with numerous methods proposed to enhance the reasoning capabilities of LLM. Chain-of-Thought (CoT) Wei et al. (2022b) and scratchpad Nye et al. (2021) encourages step-by-step reasoning by breaking down complex problems into manageable steps, while CoT's extension, self-consistency Wang et al. (2022), improves performance by sampling multiple reasoning paths and selecting the most consistent solution through majority voting. These methods have been extensively evaluated across a variety of benchmarks, including commonsense reasoning Lourie et al. (2021); Geva et al. (2021), mathematical reasoning Cobbe et al. (2021b); Hendrycks et al. (2021), symbolic reasoning Luo et al. (2023); Han et al. (2022); Patel et al. (2024); Parmar et al. (2024), logical reasoning Liu et al. (2020); Tafjord et al. (2020), and multi-modal reasoning Zellers et al. (2019a); Xiao et al. (2021); Lu et al. (2022); Handa et al. (2024). These advancements have enhanced the diversity and reliability of multi-step reasoning evaluations. However, the varying levels of difficulty across steps in most reasoning tasks pose challenges for controlled assessments, particularly in quantifying error propagation and addressing context limitations. Ensuring consistent difficulty across steps is therefore crucial for transparent and reliable evaluation of the reasoning capabilities of large language models, which remains significantly underexplored.

## 2.3 RULE SYSTEM

Complexity emerges from simple rules. Modern scientific subjects such as mathematics, physics, and chemistry are generally founded on rule-based systems. These rely on axioms Suppes (2012),

theorems Russell (2020), and laws Feynman (1963) to guide reasoning and model phenomena. Arithmetic is the most basic example, providing core operations and relations that support more advanced structures. In physics, Newton's laws of motion describe how forces affect bodies, forming the basis of classical mechanics. Cellular automata illustrate how simple local rules can give rise to complex global behavior Conway et al. (1970). Rule system is also essential in program verification and automated reasoning, facilitating formal proofs through Lean theorem prover Moura & Ullrich (2021), for example. We define the ability of an LLM to faithfully execute rules step by step as computational reasoning. Based on this definition, we propose TMBench, a rule-based system grounded in Turing machines. In TMBench, LLM is prompted to output the result of each rule execution step, which is then evaluated against ground-truth trajectories for assessment. Accurate intermediate steps are essential to the integrity of the rule-based system.

## 3 TURING MACHINE BENCH

### 3.1 MOTIVATION

Reasoning is one of the most important abilities of LLMs and has gain significant attention recently. Current reasoning benchmarks focus on the final results and depend on domain-specific knowledge. For instance, the AIME benchmark, used to evaluate mathematical reasoning abilities, relies on the mastery of mathematical knowledge. Furthermore, there is a risk that models might tailor their responses to specific benchmark metrics, thereby inflating their performance. From an intuitive perspective, reasoning is the process of **selecting** appropriate rules and **applying** them faithfully step by step. We refer to this capability as *computational reasoning*. Computational reasoning is the ability to systematically select and accurately apply rules, ensuring that each step is transparent, verifiable, and grounded within the given rule system. In contrast to logical reasoning, which primarily focuses on the validity of conclusions, computational reasoning emphasizes the faithfulness and traceability of the entire reasoning process. Therefore, we propose an evaluation framework to evaluate computational reasoning by Universal Turing Machine Simulation.

### 3.2 TURING MACHINE

**Turing Machine (TM)** is a foundational model of computation Turing et al. (1936), defining algorithmic processes and computational limits. Formally, a TM is represented as $\mathcal{T} = (Q, \Gamma, \delta, q_0, H)$, where $Q$ is the set of states, $\Gamma$ the tape alphabet, $\delta : Q \times \Gamma \to Q \times \Gamma \times \{L, R\}$ the transition function, $q_0$ the initial state, and $H$ the set of halting states. A TM manipulates symbols on an infinite tape using a read-write head, guided by a transition function $\delta$, and executes computations step by step until it reaches a halting state. It serves as a fundamental abstraction for formalizing decidability and complexity classes. Beyond theoretical significance, TM are valuable for modeling symbolic reasoning and structured inference. In this work, we leverage Turing Machine principles to construct a rigorous multi-step benchmark, evaluating LLMs' ability to perform structured computations which is vital for reasoning.

**Universal Turing Machine (UTM)** is a theory model of computation, capable of simulating the behavior of any other Turing machine. As a foundation of computability theory, the UTM formalizes the notion of computational universality and provides the theoretical foundation for the Church-Turing thesis, which asserts that any effectively computable function can be executed by a Turing Machine. By encoding both a machine's description and its input on its tape, a UTM demonstrates that a single device can emulate any computational process, establishing the basis for general-purpose computation and modern computing architectures.

To facilitate theoretical analysis while preserving computational expressiveness, we adopt the $m$-Tag System, a simple and recognized model. The $m$-Tag System has been rigorously proven to be Turing-complete for $m > 1$ Wang et al. (1971); Cocke & Minsky (1964), making it a suitable abstraction for modeling universal computation within a structured and analyzable framework.

**m-Tag System** is a formal computational model introduced in Post (1943) as a simplified yet computationally equivalent variant of the Universal Turing Machine. It operates on a queue of symbols, iteratively applying production rules to modify the sequence. A tag system is formally described by a triplet $(m, A, P)$, where:

- $m$ is the deletion number, specifying how many symbols are removed from the head of queue per step.

- $A$ is a finite alphabet of symbols, from which queue elements are drawn.

- $P$ is a set of production rules mapping each $x \in A$ to a corresponding word $P(x)$, which is appended to tail of the queue.

The single-step process is formally defined as follows:

$$\text{Step} : \underbrace{x_1\, x_2 \ldots x_m\, X}_{\text{Read}} \longrightarrow \underbrace{x_1 x_2 \ldots x_m}_{\text{Delete}} X \underbrace{P(x_1)}_{\text{Write}}, \tag{1}$$

where new symbols $P(x_1)$ are appends to the tail of queue generated based on the head symbol $x_1$ and production rule. Simultaneously, $m$ symbols are deleted from the head of the queue. This process resembles the next-token prediction mechanism employed by LLMs.

Tag systems are proven Turing completeness when $m > 1$, making them the minimal yet effective computational models, see Section A for complete certification. An example of a 2-tag system simulation is provided in Table 1, illustrating the iterative process of reading, writing, and deletion until a halting condition is reached.

Table 1: Example of 2-tag systems simulation. At each step, the head symbol of the queue is read, and new symbols are appended to the tail based on production rules. The first two symbols are then removed. The system halts when the queue contains fewer than two symbols.

| **Alphabet** | {A, B, C, D, E} | {1, 2, 3, 4, 5} | {@, #, \$, %, &} |
|---|---|---|---|
| **Init** | [B A E E C] | [5 2 3 2] | [\$ @ @ #] |
| **P-Rules** | A : E D A B C | 1 : 5 5 2 | @ : % \$ # \$ |
| | B : D | 2 : 4 2 5 1 3 | # : & |
| | C : E E E D D | 3 : 4 3 1 | \$ : & |
| | D : B C | 4 : 3 4 | % : # # % % |
| | E : D | 5 : 3 | & : % # & \$ |
| **Steps** | 0. [B A E E C] (Init) | 0. [5 2 3 2] (Init) | 0. [\$ @ @ #] (Init) |
| | 1. [B̶ A̶ E E C D] | 1. [5̶ 2̶ 3 2 3] | 1. [\$̶ @̶ @ # &] |
| | 2. [E̶ E̶ C D D] | 2. [3̶ 2̶ 3 4 3 1] | 2. [@̶ # & % \$ # \$] |
| | 3. [C̶ D̶ D E E E D D] | 3. [3̶ 4̶ 3 1 4 3 1] | 3. [&̶ %̶ \$#%#&\$] |
| | ... | ... | ... |
| | 16. [B̶ C̶ D D] | 28. [5̶1̶ 3343455243134552343] | 28. [#̶ &̶ \$%#&\$&] |
| | 17. [D̶ D̶ B C] | 29. [3̶3̶ 43455243134552343431] | 29. [\$̶ %̶ # & \$ & &] |
| | 18. [B̶ C̶ D] (Halt) | 30. [4̶3̶ 455243134552343134] | 30. [#̶ &̶ \$ & & &] |

## 3.3 EVALUATION METRICS

To comprehensively evaluate the multi-step instruction following capability of LLMs in the reasoning process, we define three key metrics: Step Accuracy, Step-Weighted Accuracy, and Pass Rate.

**Step Accuracy.** It quantifies the proportion of correctly predicted queues at a given step in the reasoning process, providing a fine-grained evaluation of a model's stepwise performance. It is particularly valuable for multi-step reasoning tasks, where errors can accumulate and propagate, significantly impacting overall accuracy. The accuracy at step $i$ is defined as:

$$\text{ACC(i)} = \frac{N_{\text{correct}}(i)}{N_{\text{total}}(i)}, \tag{2}$$

where $N_{\text{correct}}(i)$ and $N_{\text{total}}(i)$ denote the number of correct predictions and the total number of predictions at step $i$, respectively.

**Step-Weighted Accuracy (SWA)**   To evaluate model performance across a sequence of reasoning steps with an emphasis on later steps, we define the *Step-Weighted Accuracy* as a weighted average of per-step accuracies:

$$\text{SWA} = \frac{1}{\sum_{i=1}^{T} w_i} \sum_{i=1}^{T} w_i \cdot \text{ACC}(i), \tag{3}$$

where $w_i$ is the weight assigned to step $i$. Setting $w_i = 1$ corresponds to uniform weighting across all steps, while $w_i = i$ linearly increases the emphasis on later steps, thereby prioritizing accurate performance in deeper stages of reasoning. We evaluate performance using Step-Weighted Accuracy (SWA) under both uniform (Uni.) and linear (Lin.) weighting schemes.

**Pass Rate.**   The pass rate quantifies the likelihood of successfully completing a given process without errors before termination or reaching the maximum number of allowed steps. This metric focuses more on the final outcome, whereas the previous metrics emphasize the process.

## 4  EXPERIMENTS AND RESULTS

In this section, we present a series of experiments conducted on TMBench using a diverse set of models and analytical approaches. The experiments and their results are outlined as follows. First, we introduce the experimental setup including datasets and LLMs. Next, we evaluate a broad range of recent open-source LLMs, spanning from 0.6B to 671B parameters, alongside proprietary models such as Gemini, Grok, and Claude. We then assess the correlation between the TMBench Pass Rate and real-world benchmarks. Finally, we conduct ablation studies including unbounded-step, temperature, alphabet, and difficulty ablations to analyze TMBench's characteristics, which inevitably fail with increasing steps due to its autoregressive nature, remain robust across different alphabets, and allow continuous and scalable control of task difficulty.

### 4.1  EXPERIMENTAL SETUP

We provide a brief overview of the experimental setup, including the datasets and the latest large language models evaluated. Detailed information is provided in the released code and data.

**Datasets.** Based on the dataset design methodology introduced in Section 3, we sample 100 instances of m-tag systems with an alphabet size of 5, where m = 2. For each system, the rule lengths range from 1 to 5, and the initial string lengths range from 2 to 9. The maximum simulation length is 30, with 11 cases halting early upon meeting the termination condition. Experiments under other parameter settings can be found in Section 4.4. In addition, we evaluate model performance on several established benchmarks, including AIME2024 MAA (2024), MATH500 Lightman et al. (2023), GPQA Diamond Rein et al. (2024), and MMLU Pro Wang et al. (2024b). Further details can be found in the Appendix.

**LLMs.** We evaluate a diverse set of state-of-the-art LLMs, including open-source models ranging from 0.6B to 671B parameters, as well as proprietary API-only models, covering various model families and architectures to ensure a comprehensive analysis. Our selection includes the instruct-tuned versions of the LLaMA models, specifically the 1B, 8B, and 70B variants, released by the LLaMA team Dubey et al. (2024). From the Qwen family, we evaluate both the instruct and preview versions of the 0.6B, 1.7B, 4B, 8B, 14B, 32B, and 72B models Yang et al. (2024), along with the language model component of the QVQ-72B-Preview multimodal system Wang et al. (2024a); Team (2024), as well as two MoE models: Qwen3-30B-A3B and Qwen3-235B-A22B. For the Gemma family, we include both Gemma3-12B and Gemma3-27B Team et al. (2025). We also incorporate the QwQ-32B-Preview and Sky-T1-32B-Preview model Team (2025). For proprietary models, including Gemini-2.5-ProDeepMind (2025), Grok-3 xAI (2025), Claude-3.7-Sonnet-Thinking Anthropic (2025), Doubao-1.5-Pro Doubao (2025), and GPT-4.1 OpenAI (2024). For efficient and high-throughput inference, we leverage the Transformers library Wolf et al. (2020) alongside vLLM Kwon et al. (2023), which provides optimized execution for large-scale model evaluations. Unless otherwise specified, we employ a greedy decoding strategy and set the maximum generation length to 16384 tokens.

Table 2: Performance of large language models across different scales (0.6B to 605B) and API-only models, evaluated on TMBench. Asterisks (*) denote widely recognized reasoning benchmarks.

| Model | SWA (Uni.) | SWA (Lin.) | Pass Rate | AIME* | MATH* | GPQA* |
|---|---|---|---|---|---|---|
| **0.6B - 8B** | | | | | | |
| Qwen3-0.6B | 0.9 | 0.1 | 0 | 6.7 | 65.0 | 21.7 |
| Llama-3.2-1B | 3.3 | 0.3 | 0 | 0 | 18.4 | 12.6 |
| Qwen3-1.7B | 1.5 | 1.2 | 1 | 26.7 | 86.4 | 32.8 |
| Qwen3-4B | 7.4 | 3.5 | 6 | 56.7 | 91.6 | 49.0 |
| Llama-3.1-8B | 4.5 | 0.4 | 1 | 0 | 46.4 | 21.2 |
| Qwen3-8B | 7.2 | 4.4 | 8 | 60.0 | 91.4 | 56.6 |
| **12B+** | | | | | | |
| Gemma-3-12B | 7.7 | 1.6 | 0 | 13.3 | 83.2 | 41.9 |
| Qwen2.5-14B | 10.3 | 2.6 | 0 | 16.7 | 79.2 | 43.9 |
| R1-Distill-Qwen-14B | 29.5 | 18.0 | 10 | 60.0 | 87.0 | 53.5 |
| Qwen3-14B | 13.8 | 9.3 | 2 | 76.7 | 93.0 | 64.1 |
| **27B+** | | | | | | |
| Gemma-3-27B | 26.1 | 12.2 | 5 | 40.0 | 88.8 | 46.0 |
| Qwen2.5-32B | 18.2 | 6.0 | 7 | 10.0 | 80.6 | 48.0 |
| Sky-T1-32B-Preview | 20.9 | 9.1 | 8 | 40.0 | 86.2 | 51.0 |
| QwQ-32B-Preview | 19.8 | 9.8 | 4 | 53.3 | 90.4 | 62.1 |
| R1-Distill-Qwen-32B | 33.5 | 22.5 | 10 | 72.6 | 94.3 | 62.1 |
| Qwen3-32B | 10.5 | 8.3 | 7 | 70.0 | 94.8 | 63.6 |
| Qwen3-30B-A3B | 56.7 | 46.8 | 16 | 80.4 | 95.9 | 65.8 |
| **70B+** | | | | | | |
| Llama-3.3-70B | 45.2 | 34.3 | 12 | 20.0 | 75.8 | 43.9 |
| QVQ-72B-Preview | 14.4 | 6.4 | 3 | 33.3 | 82.8 | 52.5 |
| Qwen2.5-Math-72B | 42.1 | 26.6 | 15 | 20.0 | 85.2 | 46.0 |
| Qwen2.5-72B | 45.6 | 29.6 | 19 | 13.3 | 82.8 | 45.5 |
| Llama-4-Scout | 7.7 | 1.2 | 1 | 28.3 | 84.4 | 58.7 |
| Llama-4-Maverick | 39.1 | 27.8 | 19 | 39.0 | 88.9 | 67.1 |
| DeepSeek-V3 | 87.4 | 84.9 | 82 | 52.0 | 94.2 | 65.5 |
| DeepSeek-R1 | 72.2 | 63.8 | 45 | 79.8 | 97.3 | 71.5 |
| Qwen3-235B-A22B | 45.2 | 32.6 | 22 | 85.7 | 93.0 | 70.0 |
| **API only** | | | | | | |
| Qwen-2.5-Max | 27.0 | 11.8 | 7 | 23.3 | 83.5 | 58.7 |
| OpenAI-O1-mini | 37.0 | 21.6 | 11 | 63.6 | 90.0 | 60.0 |
| Gemini-1.5-Pro | 40.3 | 18.4 | 10 | 20.0 | 84.2 | 56.6 |
| OpenAI-O3-mini | 37.1 | 30.5 | 7 | 87.3 | 98.5 | 79.7 |
| GPT-4.1 | 58.7 | 45.7 | 26 | 48.1 | 91.3 | 66.3 |
| Doubao-1.5-Pro | 76.9 | 69.1 | 54 | 33.3 | 88.6 | 65.0 |
| Claude-3.7-Sonnet | 85.1 | 78.1 | 69 | 61.3 | 96.2 | 78.2 |
| Grok-3-Beta | 94.6 | 92.4 | 86 | 83.9 | 94.8 | 80.2 |
| Gemini-2.5-Pro | 96.6 | 96.2 | 94 | 92.0 | 98.9 | 84.0 |

## 4.2 Evaluation of Computational Reasoning

We evaluate the computation reasoning capabilities of the latest LLMs by analyzing their performance across three metrics: *SWA Uniform*, *SWA Linear*, *Pass Rate*. These metrics offer a comprehensive evaluation of the models' ability to faithfully execute multi-step reasoning. Detailed results are summarized in Table 2, and the accuracy–step curves are shown in Figure 1. We observe that Gemini-2.5-Pro exhibits robust accuracy in computational reasoning, achieving over 90% accuracy at 30 steps, which suggests an emergent ability to simulate Turing Machine. It is found that models smaller than 4B struggle with even the first step. This observation further supports the phenomenon of **emergence** Wei et al. (2022a). The *QVQ* multimodal model exhibits a notable decline, potentially attributable to its multimodal training process.

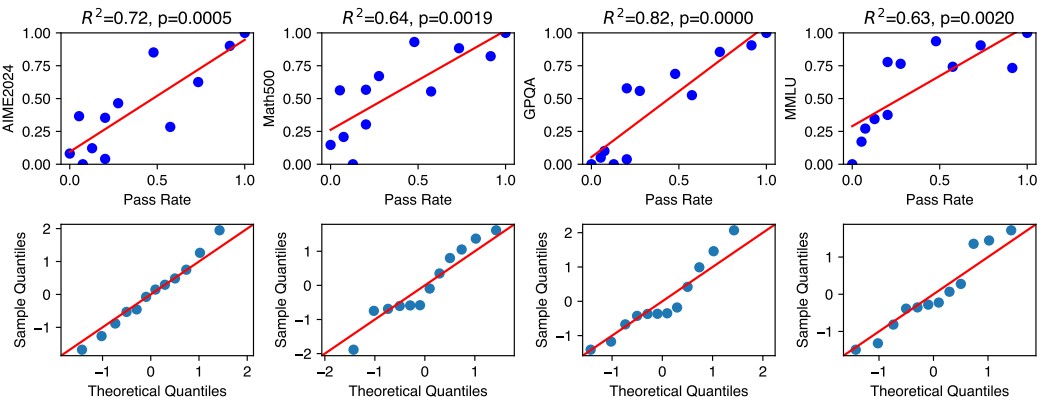

Figure 2: Correlation between TMBench Pass Rate (computational reasoning) and established benchmarks (AIME2024, MATH500, GPQA Diamond and MMLU Pro), where both metrics are min-max normalized, across 12 leading LLMs. Top: Scatter plots with linear fit between TMBench and each benchmark. Bottom: Q-Q plots of regression residuals.

**Relationship between computational reasoning and long-context.** Figure 1 shows that for models larger than 14B, such as DeepSeek-R1, accuracy decreases linearly as the number of simulated steps grows. This implies a stable per-step error probability, i.e., the difference remains constant with increasing context length. This indicates that for models larger than 14B, as the context increases, their reasoning ability does not decline and the error rate remains stable.

### 4.3 CORRELATION WITH ESTABLISHED BENCHMARKS

To investigate the relationship between computational reasoning and real-task performance, we analyzed correlations between TMBench Pass Rate and established benchmarks (AIME2024, MATH500, GPQA Diamond, MMLU Pro) across 12 leading LLMs, as shown in Figure 2. The evaluated models include Gemini-2.5-Pro, Grok-3-beta, Claude-3-7-Sonnet, Doubao-1-5-Pro, DeepSeek-R1, GPT-4.1, Llama-4-Maverick, Qwen2.5-72B, Llama-3.3-70B, Qwen2.5-32B, Gemma-3-27B, and Qwen2.5-14B. The results reveal statistically significant correlations ($p < 0.05$), confirming computational reasoning as a predictor of real-world performance. The correlation strength follows the order: GPQA > AIME2024 > MATH500 > MMLU, indicating that tasks requiring deeper reasoning align more closely with TMBench. In contrast, knowledge-heavy benchmarks like MMLU show weaker correlation. The Q-Q plots in the bottom panels suggest residuals are approximately normal, supporting regression assumptions. Furthermore, the average score across AIME2024, MATH500, and GPQA Diamond yields a Pearson correlation coefficient of 0.882 with TMBench (see Figure 5), underscoring the strong link between advanced reasoning and computational ability.

### 4.4 ABLATION STUDY

**Unbounded-Step Ablation.** Gemini-2.5-pro demonstrates strong instruction-following capabilities, achieving a 94% pass rate under a 30-step constraint. To further analyze the upper bound of its instruction-following ability, we conducted an unbounded-step evaluation. Using rejection sampling, we selected 10 samples and ran each for up to 1000 steps. Through budget forcing Muennighoff et al. (2025), Gemini was prompted to continue generating until failure with maximum token limit of 1,048,576 tokens. The earliest failure occurred at step 16, and the latest at step 683, as shown in Figure 3a. As an autoregressive model, Gemini inevitably fails with increasing steps due to its statistical nature, underscoring Gemini's computational limits.

**Temperature ablation.** To examine the effect of temperature on model performance, we conduct an ablation study with temperature values ranging from 0 to 3.0, while setting the top-p value to 0.8, as shown in Figure 3b. Performance remains stable between 0 and 1, but degrades at higher temperatures.

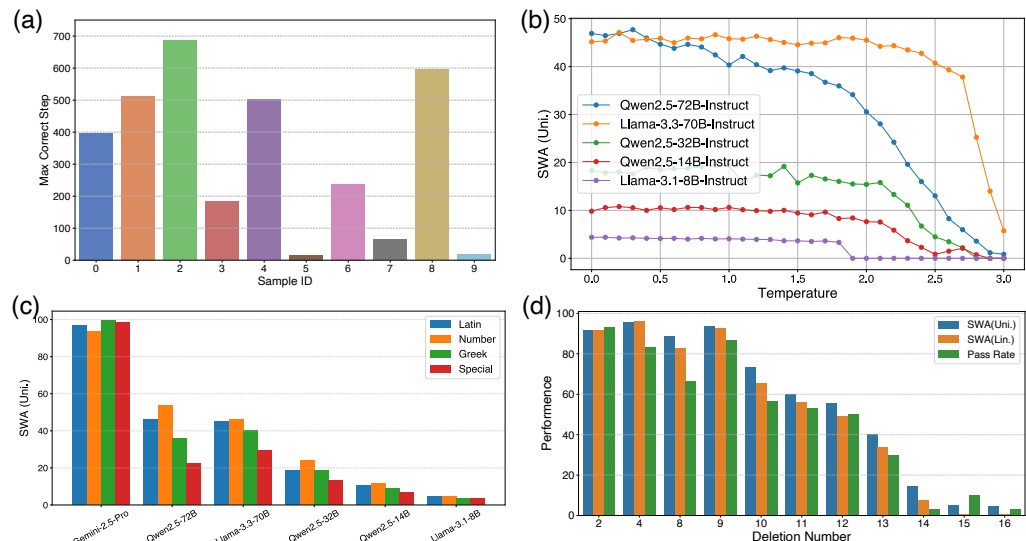

Figure 3: Illustration of ablation results. (a) Maximum correct step achieved by Gemini under unbounded-step execution. (b) Impact of decoding temperature on performance. (c) Effect of alphabet types (Roman, Number, Greek, and Special) on model performance. (d) Task difficulty ablation by varying the number of deletions.

**Alphabet ablation.** To evaluate the robustness of large language models (LLMs) across diverse alphabet types, we conducted evaluations using four distinct character sets: **Roman letters** (e.g., *a, b, c, d, e*), **Numerals** (e.g., *1, 2, 3, 4, 5*), **Greek letters** (e.g., $\alpha, \beta, \gamma, \delta, \epsilon$), and **Special characters** (e.g., @, #, $, %, &). As illustrated in Figure Figure 3c, while most models exhibit performance fluctuations depending on the character set, Gemini-2.5-Pro consistently achieves high accuracy across all categories, including the *Special characters*, which typically present greater challenges. This robustness indicates that Gemini-2.5-Pro relies on underlying reasoning mechanisms rather than superficial statistical correlations.

**Difficulty Ablation.** For TMBench, Gemini-2.5-pro achieves a 94% pass rate within 30 steps, with successful simulations extending up to 686 steps, demonstrating its strong computational reasoning capabilities. To further assess this ability, we perform a difficulty ablation study with varying values of $m$, as shown in Figure 3d. The model maintains stable performance when the deletion number $m$ is between 2 and 9, but performance declines steadily beyond $m = 10$, approaching zero after $m = 15$. This demonstrates that varying $m$ produces a smooth difficulty gradient, underscoring TMBench's scalability and effectiveness as a benchmarking tool.

## 5 CONCLUSION

We propose computational reasoning as a fundamental ability of LLMs—the capacity to strictly follow rules and manage internal states for multi-step processes, independent of domain knowledge. To evaluate this, we introduce **Turing Machine Bench (TMBench)**, a benchmark based on m-Tag System Simulation. TMBench is designed with four key attributes: (1) self-contained, knowledge-independent reasoning; (2) interpretable, minimal multi-step structure; (3) adjustable task difficulty; and (4) computational generality grounded in Turing completeness. We evaluate state-of-the-art open-source models (0.6B–671B) and proprietary models such as Gemini and Grok. Results show that TMBench captures computational reasoning effectively, revealing clear multi-step performance curves. Importantly, TMBench aligns well with real-world reasoning benchmarks (e.g., AIME, GPQA), more so than with knowledge-heavy tasks like MMLU. Ablation studies (step limits, temperature, alphabet, difficulty variations, and SFT) further demonstrate TMBench's robustness. For example, Gemini shows degradation with longer sequences due to its autoregressive nature, yet remains stable with symbol variations—suggesting reliance on true reasoning mechanisms rather than superficial correlations.

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

# A  Theory.

In this section, we demonstrate that the m-tag (m¿1) system constitutes a universal Turing machine, following the construction presented in Cocke & Minsky (1964), with minor corrections.

## A.1  Representation of a Turing Machine

We begin by introducing the definition and step operation for the Turing Machine. Consider a Turing Machine with a binary alphabet $\Gamma = \{0, 1\}$, where each state $Q_i$ is defined by a quadruple $(S_i, D_i, Q_{i0}, Q_{i1})$, corresponding to the transitions for the symbols 0 and 1. A single step of the Turing Machine, when in state $Q_i$, is performed as follows:

1. Write the symbol $S_i$ onto the current tape cell.
2. Move the tape head one cell in direction $D_i$, where $D_i \in \{L, R\}$.
3. Read the symbol $S'$ from the cell currently.
4. Transition to the next state based on the symbol read $S'$:
   - If $S' = 0$, the next state is $Q_{i0}$.
   - If $S' = 1$, the next state is $Q_{i1}$.

In this specific transition rule, the symbol written $S_i$ and the tape head direction $D_i$ are determined solely by the current state $Q_i$. The subsequent state transition depends on the symbol $S'$ read from the tape after the move: the machine transitions to state $Q_{i0}$ if $S' = 0$, and to state $Q_{i1}$ if $S' = 1$.

An instantaneous description of the machine with state $Q_i$ at any given step is represented as:

$$\cdots a_3 a_2 a_1 a_0 \alpha b_0 b_1 b_2 b_3 \cdots$$

where $Q_i$ is the state after reading $\alpha$. The complete instantaneous description can be described by a *triplet*:

$$(Q, M, N) = (Q_i, \sum_{i=0}^{\infty} a_i 2^i, \sum_{i=0}^{\infty} b_i 2^i)$$

where $M$ is the string of symbols to the left of $\alpha$, $N$ is the string of symbols to the right of $\alpha$. The symbol $\alpha$ is the current symbol being read, which will soon be replaced by $S_i$ depending on the transition rule.

For a rightward movement, the next state $Q'$ depends on the value of $S_i$. If $S_i = 0$, then $Q' = Q_{i0}$, and if $S_i = 1$, then $Q' = Q_{i1}$.

The updating of the strings follows the rules:

$$M \leftarrow 2M + S_i$$

$$N \leftarrow \left\lfloor \frac{N}{2} \right\rfloor$$

In the case of a rightward movement, only the values of $M$ and $N$ are swapped, due to the symmetry of the process. For the sake of simplicity, we assume here that the focus is on the rightward move, without loss of generality.

## A.2  Equivalent Construction with a Tag System

Given the instantaneous description of the Turing Machine as a triplet $(Q_i, M, N)$, we can construct a corresponding string using the following form:

$$A_i x_i (\alpha_i x_i)^M B_i x_i (\beta_i x_i)^N$$

For simplicity and clarity, we will omit the subscript $i$ in the notation and rewrite it as:

$$Ax(\alpha x)^M Bx(\beta x)^N$$

STEP 0 (INITIAL)

$$Ax(\alpha x)^M Bx(\beta x)^N \tag{4}$$

STEP 1

Rules:

$$A \to \begin{cases} Cx, & \text{if } S = 0 \\ Cxcx, & \text{if } S = 1 \end{cases}, \quad \alpha \to cxcx$$

Updated:

$$Bx(\beta x)^N Cx(cx)^{M'} \tag{5}$$

where

$$M' = \begin{cases} 2M, & \text{if } S = 0 \\ 2M + 1, & \text{if } S = 1 \end{cases}$$

STEP 2

Rules:

$$B \to S, \quad \beta \to s$$

Updated:

$$Cx(cx)^{M'} S(s)^N \tag{6}$$

STEP 3

Rules:

$$C \to D_1 D_0, \quad c \to d_1 d_0$$

Updated:

$$S(s)^N D_1 D_0 (d_1 d_0)^{M'} \tag{7}$$

STEP 4

Rules:

$$S \to T_1 T_0, \quad s \to t_1 t_0$$

Based on the parity of $N$, the updated string takes different forms:

If $N$ is odd:

$$D_1 D_0 (d_1 d_0)^{M'} T_1 T_0 (t_1 t_0)^{\frac{N-1}{2}},$$

If $N$ is even:

$$D_0 (d_1 d_0)^{M'} T_1 T_0 (t_1 t_0)^{\frac{N}{2}}.$$

STEP 5

If $N$ is odd:

$$D_1 \to A_1 x_1, \quad d_1 \to a_1 x_1$$

Then, the resulting string is:

$$T_1 T_0 (t_1 t_0)^{\frac{N-1}{2}} A_1 x_1 (\alpha_1 x_1)^{M'} \tag{8}$$

If $N$ is even:

$$D_0 \to x_0 A_0 x_0, \quad d_0 \to a_0 x_0$$

Then, the resulting string is:

$$T_0 (t_1 t_0)^{\frac{N}{2}} x_0 A_0 x_0 (\alpha_0 x_0)^{M'} \tag{9}$$

STEP 6 (FINAL)

If $N$ is odd:
$$T_1 \to B_1 x_1, \quad t_1 \to \beta_1 x_1$$
Then,
$$A_1 x_1 (\alpha_1 x_1)^{M'} B_1 x_1 (\beta_1 x_1)^{\frac{N-1}{2}} \tag{10}$$

If $N$ is even:
$$T_0 \to B_0 x_0, \quad t_0 \to \beta_0 x_0$$
Then,
$$A_0 x_0 (\alpha_0 x_0)^{M'} B_0 x_0 (\beta_0 x_0)^{\frac{N}{2}} \tag{11}$$

This completes the proof that a single computation step of a Turing machine is successfully simulated by a sequence of transitions in the 2-tag system.

## B   EXPERIMENT DETAILS

### B.1   DATASET DETAILS

**AIME2024** MAA (2024) is a collection of problems from the 2024 American Invitational Mathematics Examination. It contains 30 high-school level mathematics problems requiring creative problem solving and deep mathematical insight. These problems are typically used to select top students for the USA Mathematical Olympiad. The dataset emphasizes algebraic manipulation, mathematical reasoning, and non-routine problem solving.

**Math500** Lightman et al. (2023) is a representative benchmark consisting of 500 mathematics problems, sampled at random from the MATHHendrycks et al. (2021) evaluation dataset by OpenAI. It includes problems of difficulty across topics such as algebra, geometry, and number theory.

**GPQA Diamond** Rein et al. (2024) is a high-difficulty subset of Graduate-Level Google-Proof Q&A Benchmark (GPQA), consisting of 198 multiple-choice questions in biology, physics, and chemistry. It includes only questions where both experts answer correctly and the majority of non-experts answer incorrectly to ensure highest quality.

**MMLU Pro** Wang et al. (2024b) is a professional-level extension of the original Massive Multitask Language Understanding (MMLU) benchmark Hendrycks et al. (2020), comprising expert-level questions across 57 domains, including mathematics, history, computer science, law, medicine, engineering, the natural sciences, and more. It enhances the original benchmark by introducing more challenging, reasoning-intensive questions, expanding answer choices from four to ten, and removing trivial or noisy items.

**TMBench** summarizes the distributions of Step Length, Rule Length, and Initial Length as shown in the following tables.

Table 3: Distribution of Step Length

| Step Range | 1–6 | 7–12 | 13–18 | 19–24 | 25–30 |
|:---:|:---:|:---:|:---:|:---:|:---:|
| Avg. | 7.61 | 13.60 | 19.41 | 25.11 | 34.35 |

Table 4: Distribution of Rule Length

| Range | 1.40–2.08 | 2.08–2.76 | 2.76–3.44 | 3.44–4.12 | 4.12–4.80 |
|:---:|:---:|:---:|:---:|:---:|:---:|
| Percentage | 0.08 | 0.25 | 0.39 | 0.23 | 0.05 |

### B.2   IMPLEMENTATION DETAILS

**Environment.** Our method is implemented with Python 3.10.16, CUDA 12.4, PyTorch 2.6.0 and vLLM 0.8.5. The required libraries are specified in the `requirements.txt` file provided in the repository. The experiments are performed on a machine with 96 vCPUs (2.90 GHz) from Intel Xeon processors, eight NVIDIA A100-80GB GPUs, and 1024 GB of RAM.

Table 5: Distribution of Initial Length

| Range | 2.00–3.40 | 3.40–4.80 | 4.80–6.20 | 6.20–7.60 | 7.60–9.00 |
|---|---|---|---|---|---|
| Percentage | 0.18 | 0.32 | 0.28 | 0.14 | 0.08 |

**Inference Configuration.** For API-only models, we use their official APIs and set a maximum token length limit accordingly. For open-source models that exceed the GPU memory limits of a single machine, such as LLaMA 4 Maverick, DeepSeek-V3, and DeepSeek-R1, we rely on APIs provided by Alibaba Cloud and DeepSeek. For all other models, including Qwen, LLaMA, and Gemma, we use vLLM as the inference backend, which is highly efficient. We apply a sampling strategy based on greedy decoding with a maximum token limit of 16,384.

**Prompt.** We adopt the following 1-shot reasoning prompt template for evaluation, where the shot example is used to guide the output format extraction.

```
<|im_start|>user
Simulate a m-tag system.  Your task is to simulate each
transition step-by-step and provide the queue's state at each
step.  Follow the rules and examples closely, and stop upon
reaching the halt condition or 30 steps.  Do not generate
additional examples or new problems.  No code.
## Rules for Simulation:
1.  In each transition, the machine performs the following
steps:
  - If the queue length is less than m, halt
  - Read the head symbol of queue
  - Append symbols to the tail based on the head symbol and the
corresponding transition rule
  - Delete m symbols from the head of the queue

2.  The machine halt if:
  - The queue's length is less than m.

## Example:
m:  2
Alphabet:  A, B, C
Init:  [B C A]
Transition rules:
A : C A C
B : A
C : B
Simulation steps:
### step 0:
- Action:  Init
- Queue State:  [B C A]

### step 1:
- Head Symbol:  B
- Action:  Append A to the end of the queue.  Remove B C from
the head.
- Queue State:  [A A]
 ...
### step 4:
- Head Symbol:  C
- Action:  Append B to the end of the queue.  Remove C B from
the head.
- Queue State:  [B] <halt>

___
## The Only Problem to Solve:
m:  {m}
Alphabet:  {ALPHABET}
Init:  {INIT}
Transition Rules:
{RULES}
Simulation steps:
<|im_end|>
<|im_start|>assistant
```

**Case.** The response generated by Gemini-2.5-Pro, showcasing its impressive capabilities in instruction following and reasoning.

```
<|im_start|>assistant
Here is the step-by-step simulation:
m:  2
Alphabet:  A, B, C, D, E
Init:  [B D D]
Transition Rules:
A : C
B : E C E C
C : B B B A
D : D B B
E : A E E E
Simulation steps:
### step 0:
- Action:  Init
- Queue State:  [B D D]
### step 1:
- Head Symbol:  B
- Action:  Append E C E C to the end of the queue.  Remove B D
 from the head.
- Queue State:  [D E C E C]
### step 2:
- Head Symbol:  D
- Action:  Append D B B to the end of the queue.  Remove D E
 from the head.
- Queue State:  [C E C D B B]
 ...
### step 18:
- Head Symbol:  E
- Action:  Append A E E E to the end of the queue.  Remove E C
 from the head.
- Queue State:  [E C A E E E A E E E A E E E A E E E A E E E A
 E E E A E E E A E E E A E E E]
### step 19:
- Head Symbol:  E
- Action:  Append A E E E to the end of the queue.  Remove E C
 from the head.
- Queue State:  [A E E E A E E E A E E E A E E E A E E E A E E
 E A E E E A E E E A E E E _ _ _ _] (Answer)
- Queue State:  [A E E E A E E E A E E E A E E E A E E E A E E
 E A E E E A E E E A E E E A E E E] (Ground Truth)
 ...
```

### B.3 MORE RESULTS

**Arithmetic tasks.**    To comprehensively evaluate mathematical reasoning, we consider a suite of challenging benchmarks, including AIME, GSM8K, MATH, GPQA, and MMLU, with results summarized in Table 6.

| Model | TMBench | GSM8K | AIME | MATH | GPQA | MMLU |
|---|---|---|---|---|---|---|
| Qwen3-14B | 30.6 | 96.3 | 76.7 | 93.0 | 64.1 | 67.5 |
| Gemma3-27B-IT | 25.8 | 94.7 | 40.0 | 88.8 | 46.0 | 66.9 |
| Qwen3-30B-A3B | 54.3 | 96.2 | 80.4 | 95.9 | 65.8 | 71.0 |
| Qwen3-32B | 46.7 | 96.5 | 70.0 | 94.8 | 63.6 | 72.7 |
| Qwen3-235B-A22B | 46.6 | 96.7 | 85.7 | 93.0 | 70.0 | 76.2 |

Table 6: Experimental results on arithmetic and reasoning benchmarks.

**Ablation on Alphabet Size.**    We observe that when other parameters are held constant, models perform worse with a small alphabet size. With a small alphabet, such as size = 2, the same symbols frequently appear in production rules. Different rules tend to look alike creating similar patterns that large language models struggle to differentiate, leading to more errors. As the alphabet size increases, each symbol appears less frequently, making the rules more distinct and diverse. This uniqueness allows the model to more easily manage the internal states to give the correct results (see Table 7).

| Alphabet Size | Qwen3-14B | Gemma3-27B-IT | Qwen3-30B-A3B | Qwen3-32B | Qwen3-235B-A22B | Avg |
|---|---|---|---|---|---|---|
| 2 | 13.2 | 8.7 | 13.9 | 20.2 | 15.4 | 14.3 |
| 5 | 30.6 | 25.8 | 54.3 | 46.7 | 46.6 | 40.8 |
| 8 | 51.1 | 31.8 | 75.5 | 56.6 | 65.5 | 56.1 |
| 12 | 64.3 | 44.5 | 80.3 | 77.0 | 70.9 | 67.4 |
| 18 | 67.8 | 52.1 | 90.4 | 86.7 | 84.8 | 76.4 |
| 26 | 72.7 | 55.8 | 90.7 | 90.1 | 83.6 | 78.6 |

Table 7: Performance comparison across different alphabet sizes.

**Ablation on Rule Length.**    The results in the  Table 8 indicate that increasing the rule length leads to higher task complexity and a corresponding decline in model performance.

| Rule Length | Qwen3-14B | Gemma3-27B-IT | Qwen3-30B-A3B | Qwen3-32B | Qwen3-235B-A22B | Avg |
|---|---|---|---|---|---|---|
| 1–3 | 41.3 | 27.9 | 63.6 | 64.0 | 63.3 | 52.0 |
| 1–4 | 38.6 | 24.4 | 58.8 | 51.4 | 53.4 | 45.3 |
| 1–5 | 30.6 | 25.8 | 54.3 | 46.7 | 46.6 | 40.8 |
| 2–6 | 28.7 | 24.9 | 51.5 | 44.0 | 46.9 | 39.2 |
| 2–7 | 22.2 | 19.6 | 49.0 | 39.0 | 41.2 | 34.2 |

Table 8: Performance comparison across different Rule Length.

**Ablation on Initial Length.**    We conducted ablations across different initial lengths and found that model performance shows no significant correlation with initial length, suggesting that initial length has minimal impact on the overall benchmark difficulty, as show in Table 9

**Impact of SFT on TMBench.**    To investigate whether TMBench primarily rewards surface-pattern learning, we performed supervised fine-tuning on synthetic traces with different alphabet sizes ($A = 2$ and $A = 5$), as well as on TEST configurations. While SFT consistently improved performance across all models, the gains remained limited compared to frontier systems such as Gemini-2.5-Pro, Claude-3, and Grok-2. Moreover, training on TEST traces did not yield substantially higher performance than training on Non-TEST traces. These results indicate that although SFT helps models capture certain superficial patterns, it is insufficient to bridge the performance gap, reinforcing

| Initial Length | Qwen3-14B | Gemma3-27B-IT | Qwen3-30B-A3B | Qwen3-32B | Qwen3-235B-A22B | Avg |
|---|---|---|---|---|---|---|
| 2–5 | 28.4 | 23.1 | 52.7 | 44.4 | 46.5 | 39.0 |
| 2–7 | 32.3 | 18.3 | 53.6 | 43.6 | 49.3 | 39.4 |
| 2–9 | 30.6 | 25.8 | 54.3 | 46.7 | 46.6 | 40.8 |
| 3–11 | 40.6 | 29.0 | 57.4 | 48.1 | 44.1 | 40.2 |
| 5–13 | 42.4 | 29.7 | 52.7 | 46.8 | 49.3 | 40.3 |

Table 9: Performance comparison across different Initial Length.

our claim that TMBench demands genuine reasoning and long-horizon state tracking rather than simple template memorization.

Table 10 reports the quantitative outcomes. Although SFT improves model performance, the magnitude of improvement is significantly smaller than in other benchmarks, where TEST-based SFT typically delivers substantial gains. The relatively limited benefits on TMBench thus underscore its difficulty and robustness against shortcut learning.

| Model | Llama-3.1-8B | Qwen3-8B | Gemma3-12B | Qwen3-14B |
|---|---|---|---|---|
| Baseline | 4.4 | 22.8 | 7.7 | 30.6 |
| SFT ($A = 2$) | 5.5 | 26.4 | 11.7 | 32.5 |
| SFT ($A = 5$) | 9.4 | 24.7 | 28.0 | 37.8 |
| SFT ($A = 5$, TEST) | 9.2 | 28.5 | 28.2 | 39.7 |

Table 10: Performance of different models on TMBench before and after SFT. All models were fine-tuned with a learning rate of $1.0 \times 10^{-4}$ for 3 epochs.

**Prompt ablation.** Since large language models (LLMs) are known to be sensitive to prompt design, we further investigated the effect of prompt variation on TMBench. In our setup, we adopted a 1-shot chain-of-thought (CoT) template to ensure that model outputs follow a step-by-step format suitable for automatic extraction using regular expressions, thereby enabling fine-grained evaluation at the step level. To assess the robustness of model rankings, we performed ablation studies by varying the number of demonstrations in the prompt (1-shot, 2-shot, 4-shot, and 8-shot). As shown in Table 11, the relative rankings of models remained consistent across different prompt settings, suggesting that TMBench performance is not overly dependent on a particular prompt configuration.

| Setting | Qwen3-14B | Gemma3-27B-IT | Qwen3-30B-A3B | Qwen3-32B | Qwen3-235B-A22B |
|---|---|---|---|---|---|
| 1-shot | 30.6 | 25.8 | 54.3 | 46.7 | 46.6 |
| 2-shot | 35.3 | 25.6 | 53.2 | 44.8 | 48.5 |
| 4-shot | 36.0 | 26.1 | 54.7 | 45.2 | 43.6 |
| 8-shot | 35.3 | 25.9 | 56.1 | 47.9 | 46.0 |

Table 11: Prompt ablation study on TMBench. Performance of different models under varying numbers of in-context demonstrations.

**Token Distribution.** We analyze the token distribution across 36 models on TMBench, as illustrated in Figure 4. A larger number of generated tokens does not necessarily correlate with better performance. While models that engage in more reasoning often produce longer outputs, this behavior does not always translate into higher accuracy. In some cases, excessive token counts are primarily due to repetitive output, reflecting the well-known repetition problem. Interestingly, state-of-the-art models such as Gemini-2.5-Pro, Grok, and Claude exhibit similar token distributions, suggesting that high-performing systems strike a balance between reasoning depth and output efficiency.

To provide a more comprehensive view, we further examine average token counts, efficiency (defined as the ratio of total tokens to pass rate), and the distribution of token lengths across different ranges. As shown in Table 12, strong models such as Gemini-2.5-Pro and Grok maintain moderate token lengths and high efficiency, whereas weaker models (e.g., Qwen3-30B-A3B, QwQ-32B-Preview, and R1-Distill-Qwen-14B) tend to produce extremely long outputs with low efficiency. These findings

indicate that TMBench not only evaluates reasoning ability but also reflects a model's capacity to generate concise and effective outputs.

| Model | Pass Rate | Avg Token | Efficiency | < 1000 | 1000–5000 | 5000–10000 | > 10000 |
|---|---|---|---|---|---|---|---|
| Gemini-2.5-Pro | 94 | 1993.6 | 2120.9 | 0.09 | 0.91 | 0 | 0 |
| Grok-3-Beta | 86 | 2182.8 | 2538.1 | 0.08 | 0.92 | 0 | 0 |
| Claude-3.7-Sonnet | 69 | 1640.3 | 2377.2 | 0.15 | 0.85 | 0 | 0 |
| Doubao-1.5-Pro | 54 | 1932.0 | 3577.8 | 0.10 | 0.90 | 0 | 0 |
| Deepseek-R1 | 45 | 1833.4 | 4074.2 | 0.10 | 0.90 | 0 | 0 |
| Qwen3-30B-A3B | 16 | 8500.8 | 53129.7 | 0.01 | 0.11 | 0.57 | 0.31 |
| QwQ-32B-Preview | 4 | 5636.8 | 140920.2 | 0.02 | 0.58 | 0.28 | 0.12 |
| R1-Distill-Qwen-14B | 10 | 10235.1 | 102350.8 | 0 | 0.08 | 0.44 | 0.48 |
| Qwen3-4B | 6 | 13695.6 | 228259.8 | 0 | 0.11 | 0.15 | 0.74 |

Table 12: Token distribution analysis on TMBench. We report pass rate, average token count, efficiency (total tokens divided by pass rate), and the proportion of outputs within different token length ranges.

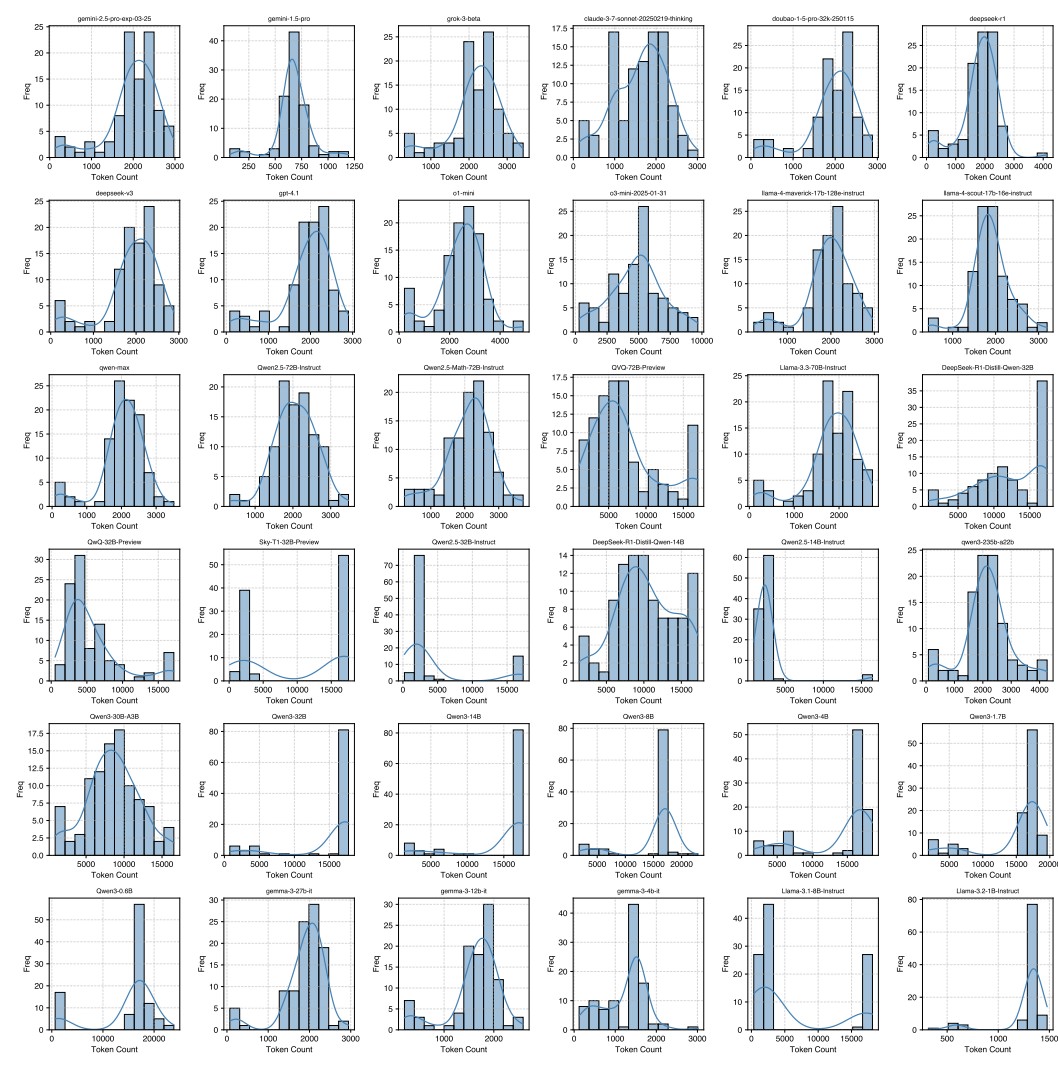

Figure 4: Illustration of token distrubition.

**Benchmark Correlation.** We compute the average score across AIME2024, MATH500, and GPQA Diamond, obtaining a Pearson correlation coefficient of 0.882 with the TMBench pass rate, as

shown in Figure 5. This strong correlation suggests that advanced reasoning performance is closely associated with computational reasoning ability.

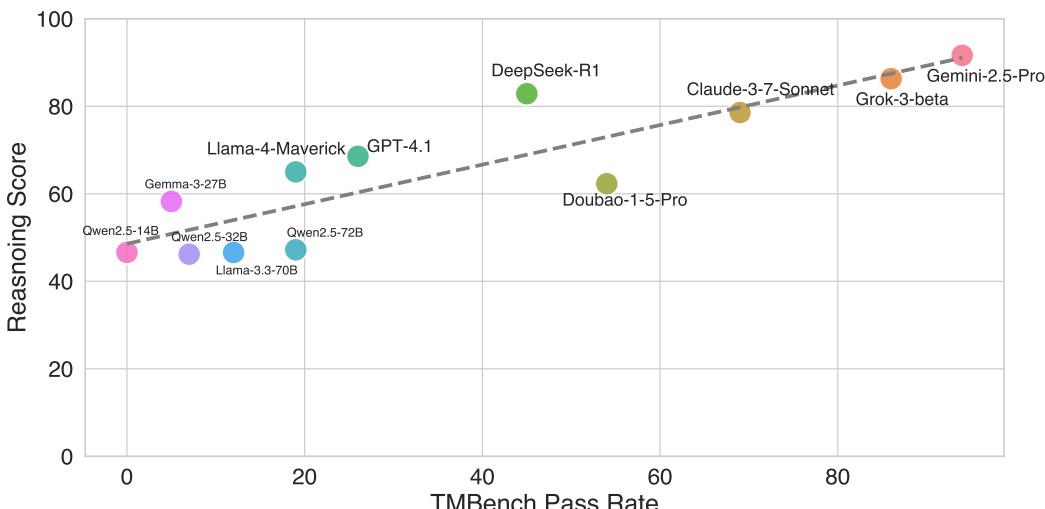

Figure 5: Correlation between TMBench Pass Rate (computational reasoning ability) and Reasoning Score (averaged across AIME2024, MATH500, and GPQA) among LLMs, with a Pearson correlation coefficient of 0.882 with p=1.49e-04, demonstrating the connection between TMBench as an abstract rule-based system simulation and real-world reasoning problem.

## C LLM IN PAPER WRITING

We use LLMs only to aid and polish writing.

