# OpenReview forum: "Computational Reasoning of Large Language Models"
_ICLR.cc/2026/Conference — Submitted to ICLR 2026_

### Official Review · Reviewer_b1Yx · 2025-10-23

**Soundness:** 3
**Presentation:** 2
**Contribution:** 2
**Rating:** 4
**Confidence:** 2

**Summary:**

The paper proposes the Turing Machine Bench (TMBench), which uses an m-Tag system to simulate a Turing machine. It uses self-consistent, knowledge-independent, multi-step rule execution to evaluate the computational reasoning capabilities of large models, and provides controllable difficulty and step-by-step metrics (SWA and Pass Rate). On a variety of open-source and closed-source models, TMBench demonstrates that top models can stably execute dozens of steps while still accumulating error with the number of steps. It also significantly correlates with high-level reasoning benchmarks such as AIME, MATH, and GPQA. These results demonstrate that TMBench can serve as a cross-domain, simple, and unified proxy for reasoning capabilities.

**Strengths:**

1.This paper proposes TMBench based on m-Tag Turing machines as a self-consistent, knowledge-independent unified reasoning evaluation framework, which solves the problem that existing benchmarks mix knowledge and reasoning and lack simple generalizable proxy tasks.
2. This article provides step-by-step interpretable curves and adjustable difficulty, and empirically proves a strong correlation with AIME/MATH/GPQA, solving the problem that multi-step reasoning processes are difficult to quantify error propagation and difficult to align evaluation with real-world tasks.

**Weaknesses:**

1. Are 100 samples too few for a general inference benchmark？ Maybe it will limit statistical robustness and extrapolation. Although each sample contains multiple steps of trajectory to increase the data points, the diversity is still limited.
2. There are some writing problems in the article：

 **line 178** “…has gain significant attention…”→“…has gained significant attention…”

**line 458** "Figure Figure 3c" --> "Figure 3c"

**line 1238** “token distrubition”-->“token distribution”

Inconsistent terminology: "computation reasoning" and "computational reasoning" are used interchangeably. It is recommended to unify them into "computational reasoning"

**Questions:**

see weaknesses

---

### Official Review · Reviewer_crJ8 · 2025-10-25

**Soundness:** 2
**Presentation:** 1
**Contribution:** 2
**Rating:** 2
**Confidence:** 4

**Summary:**

This paper presents a new benchmark for testing the in-context reasoning capabilities of large language models, using recursively enumerable languages (those accepted by a Turing Machine). They use their benchmark to evaluate 35 large language models, finding that Gemini-2.5-Pro is the best performer.

**Strengths:**

The paper has the following strengths:
* A commendably large set of language models are evaluated on the proposed benchmark.
* The methodological decisions, such as focusing on m-tag systems, are defensible, given the limited space in a conference paper.

**Weaknesses:**

I am recommending reject for this paper simply because it does not engage with the state of the art when it comes to (a) in-context learning in language models and (b) using formal languages/automata theory to study the capabilities of these models. Essentially, I believe that most of what the authors propose has already been done, with a greater degree of formal rigour. While I agree that no-one has proposed quite the same benchmark as the authors do here, I don't think that what they offer is altogether that novel. I provide references to the relevant literature below:

### In-Context Learning

The authors are essentially proposing a benchmark of what pre-trained transformers models can learn in-context, which is a live and active research area that the authors do not engage with at all:

Akyürek, E., Wang, B., Kim, Y., & Andreas, J. (2024). In-context language learning: Architectures and algorithms. arXiv preprint arXiv:2401.12973.

Coda-Forno, J., Binz, M., Akata, Z., Botvinick, M., Wang, J., & Schulz, E. (2023). Meta-in-context learning in large language models. Advances in Neural Information Processing Systems, 36, 65189-65201.

Lampinen, A. K., Chan, S. C., Singh, A. K., & Shanahan, M. (2024). The broader spectrum of in-context learning. arXiv preprint arXiv:2412.03782.

Von Oswald, J., Niklasson, E., Randazzo, E., Sacramento, J., Mordvintsev, A., Zhmoginov, A., & Vladymyrov, M. (2023, July). Transformers learn in-context by gradient descent. In International Conference on Machine Learning (pp. 35151-35174). PMLR.

Xie, S. M., Raghunathan, A., Liang, P., & Ma, T. (2021). An explanation of in-context learning as implicit bayesian inference. arXiv preprint arXiv:2111.02080.

### Formal Automata Theory

Automata theory has been used extensively to study the capabilities of language models. I encourage the authors to thoroughly review the following papers and the references that they cite:

Ackerman, J., & Cybenko, G. (2020). A survey of neural networks and formal languages. arXiv preprint arXiv:2006.01338.

Butoi, A., Khalighinejad, G., Svete, A., Valvoda, J., Cotterell, R., & DuSell, B. (2024). Training neural networks as recognizers of formal languages. arXiv preprint arXiv:2411.07107.

Borenstein, N., Svete, A., Chan, R., Valvoda, J., Nowak, F., Augenstein, I., ... & Cotterell, R. (2024). What languages are easy to language-model? a perspective from learning probabilistic regular languages. arXiv preprint arXiv:2406.04289.

Nowak, F., Svete, A., Butoi, A., & Cotterell, R. (2024). On the representational capacity of neural language models with chain-of-thought reasoning. arXiv preprint arXiv:2406.14197.

Li, J., White, J. C., Sachan, M., & Cotterell, R. (2024). A transformer with stack attention. arXiv preprint arXiv:2405.04515.

Svete, A., Chan, R. S. M., & Cotterell, R. (2024). On efficiently representing regular languages as RNNs. arXiv preprint arXiv:2402.15814.

Voudouris, K., Barron, A., Halina, M., Klein, C., & Patel, M. (2025). Exploring Major Transitions in the Evolution of Biological Cognition With Artificial Neural Networks. arXiv preprint arXiv:2509.13968.

Weiss, G., Goldberg, Y., & Yahav, E. (2018). On the practical computational power of finite precision RNNs for language recognition. arXiv preprint arXiv:1805.04908.

**Questions:**

* Do the authors expect any capability boosts from conducting post-training (e.g., supervised fine-tuning) on held-out m-tag problems? I would be interested to see how fine-tuning boosts in-context learning performance.

---

### Official Review · Reviewer_wEq4 · 2025-10-30

**Soundness:** 3
**Presentation:** 3
**Contribution:** 3
**Rating:** 6
**Confidence:** 4

**Summary:**

This paper introduces the Turing Machine Benchmark (TMBench), which evaluates the ‘computational reasoning’ capabilities of large language models by simulating simplified Turing machines (m-Tag systems). Each task requires the model to incrementally update a symbol queue according to fixed rules. The authors define granular metrics—step accuracy, step-weighted accuracy, and pass rate—to quantify correctness per step and overall success. Key findings include: accuracy exhibits a systematic decline with increasing steps, and TMBench pass rates demonstrate strong empirical correlation with standard reasoning benchmarks.

**Strengths:**

1.Introducing TMBench as a domain-agnostic proxy task appears novel. By basing it on m-Tag (a simple Turing machine variant), the authors tie LLM evaluation to well-understood computational primitives. This abstraction is a strong point, as it isolates rule-following ability from background knowledge.

2.A wide array of models (from small open models to top proprietary ones) are evaluated under consistent settings. The experiments cover not only basic performance but also varied conditions (unlimited steps, temperature changes, different alphabets, task difficulty via deletion count). This thoroughness lends credibility to the claims.

3.Finding a high Pearson correlation (≈0.882) between TMBench pass rate and averaged reasoning scores is a compelling result. It suggests that this abstract benchmark may indeed reflect “real” reasoning ability, thus strengthening the paper’s significance. The authors also check residuals (Q-Q plots) to validate regression assumptions.

**Weaknesses:**

1.TMBench tasks are highly artificial (m-Tag simulations) and quite different from natural language problems. It is unclear how well success on these tasks truly reflects broader reasoning skills beyond what is measured. The paper posits “computational generality”, but only one type of formalism is used.

2. The correlation claims are based on 12 data points (one per model), so even though p<0.05 is reported, this may not be robust. A single outlier model could sway the result. The analysis might benefit from confidence intervals or non-parametric checks.

3. The paper lacks simple baselines. For example, it would strengthen conclusions to compare LLMs against trivial strategies (like a rule-based simulator) or to random guessing. Also, only one prompting style (1-shot example) is used; variations (0-shot, few-shot with explanations) are not explored.

**Questions:**

1.How exactly is a prediction counted as “correct” at each step? Does the model have to output the entire queue state string perfectly, including brackets and commas?

2. Given only 12 models in the correlation, have you tested the robustness of the Pearson r? For example, what is the confidence interval of the correlation, or how would results change if one model is removed?

3. You evaluate up to 30 steps by default. How did you choose the 30-step limit? If tasks ran longer (or shorter), would model rankings or metric behavior change qualitatively?

4. In the SFT experiment (Sec.4.4/Table 10), how was the synthetic fine-tuning data generated? Is there a risk that models learn just the training tasks rather than true reasoning?

---

### Official Review · Reviewer_m5Wv · 2025-11-03

**Soundness:** 3
**Presentation:** 2
**Contribution:** 2
**Rating:** 4
**Confidence:** 4

**Summary:**

This paper proposes TMBench, a benchmark for evaluation the "computational reasoning" ability of LLMs in a domain-independent and universal way. The authors make an argument for why this benchmark adds value to already existing set of benchmarks assessing various reasoning strengths of LLMs. They do an extensive evaluation of various LLMs and also compare correlation between TMBench scores and scores on other, as they call it "real-world", benchmarks.

**Strengths:**

* TMBench is a novel benchmark for assessing the reasoning abilities of models in a domain-independent fashion.

* The authors show that performance on TMBench correlates with exisitng benchmarks for math reasoning, etc.

* The synthetic nature of TMBench allows one to generate instances of varying difficulty, similar to some earlier works such as on grid logic puzzles and datasets inspired by constraint satisfaction problems. This allows the method to be "future proof" in the sense that if the latest LLMs solve the current instances, one can easily generate harder ones, again challenging the models further.

* It is useful to know that changing the alphabet (Roman, numeral, Greek, etc.) doesn't really affect the performance of the tested LLMs, suggesting that they aren't solving these tasks via pattern matching and memorization.

**Weaknesses:**

* I am not convinced by the authors' pitch that this is the first benchmark that attempts to measure the computational reasoning ability of LLMs. There have been many attempts, some specifically targeting domain-agnostic assessments, such as multiple papers proposing various kinds of grid logic puzzles as benchmarks.

* While I see that being able to simulate a Turing Machine is an interesting ability to measure, the authors miss an important distinction -- namely that *simulating* a given Turing Machine is different from *learning* (from data) what the rules of a machine should be and then being able to execute that specific Turing Machine, which is how machine learning models operate. So, for example, even if LLMs do well on the TMBench benchmark, it does not mean they can learn to (or be parameterized to) solve a specific task such as math reasoning or program execution. Additionally, in some ways, specific tasks such as solving math problems are much closer to real needs than simulating a (given) abstract Turing Machine.

* The authors also don't clearly distinguish between chain-of-thought "reasoning" LLMs vs. non-reasoning LLMs. This distinction is crucial as, from a formal standpoint, it is now understood that basic transformers can only solve a subset of polynomial-time problems (namely, those that are highly parallelizable) [1,2], and that with chain-of-thought, they become universal [3,4]. The paper does add a complementary angle, namely a systematic empirical assessment of transformer-based LLMs. It would be more valuable if they could frame the discussion and empirical findings in light of what's known theoretically about these models.

* I think the title is just too generic -- it makes the paper sound much broader than what its main contribution is, which is to propose a new new benchmark (TMBench) and report LLM performance results on it. At first glance, it also makes one think that this might be the first paper addressing the computational reasoning ability of LLMs, which clearly isn't the case.

* Rule following ability of LLMs has been explored via other benchmarks as well, such as RuleTaker [5], PRover [6], and even extensions of the bAbI benchmark from many years ago. Since the author emphasize rule following ability, placing TMBench in the context of these earlier works on rule following would be appropriate.

* Some of the empirical results are confusing or confusingly presented. E.g., at the end of section 4.2, it's noted that about Fig 1 that models larger than 14B size drop accuracy linearly as the number of simulated steps grows. However, (a) of the 12 models plotted in fig 1, only 1 model has size under 14B, and (b) unless the other plots are extended much farther to the right (till the accuracy drops close to zero), it isn't really possible to tell if the drop is linear or not (e.g., the drop for Qwen2.5-32B looks linear for the first 10 steps, but after that we see it isn't really linear). As another example, the choice of a column plot for Fig 3(a) is odd; I would have chosen some way to illustrate the distribution (e.g., with max correct steps on the x-axis, or a sorted table).

* The dataset itself is very succinctly described (in one paragraph in section 4.1). It's unclear whether the generated problems are hard or easy, whether they correspond to any naturally interesting problems or not, etc. As such, it's not totally clear how the results should be interpreted -- what does it mean to be able to do well (or do poorly) on these 100 problem instances?

[1] The Parallelism Tradeoff: Limitations of Log-Precision Transformers, Merrill et al., 2022

[2] Transformers in Uniform TC0, Chiang, 2024

[3] The Expressive Power of Transformers with Chain of Thought, Merrill et al., 2024

[4] Chain of Thought Empowers Transformers to Solve Inherently Serial Problems, Li et al., 2024

[5] Transformers as Soft Reasoners over Language, Clark et al., 2020

[6] PRover: Proof Generation for Interpretable Reasoning over Rules, Saha et al., 2020

**Questions:**

I don't have specific questions, but there are some included in the "weaknesses" section that the authors might want to address.

---

### Meta-Review · Area_Chair_njLr · 2026-01-07

**Summary:**

This submission proposes TMBench, a Turing-machine–inspired benchmark (m-tag systems) intended to measure “computational reasoning,” i.e., strict rule following + internal state tracking across multiple steps, with controllable difficulty and step-wise metrics. Reviewers generally agree the benchmark is well-motivated as a domain-agnostic probe, the experiments are broad (many models, multiple settings), and the paper provides useful diagnostics (accuracy-vs-steps curves) and correlation evidence with established reasoning benchmarks. However, many concerns center on (i) overstated novelty relative to substantial prior work on formal languages and rule-following benchmarks, (ii) limited methodological clarity and baseline choices, (iii) statistical fragility of correlation claims (small N models), (iv) presentation issues (figures, dataset description depth, and paper framing/title). Based on the mixed scores and outstanding concerns about novelty framing and empirical/statistical rigor, my suggested decision leans Reject.

**Reviewer Concerns:**

Addressed
Writing/terminology issues
Some experimental breadth

Perhaps not addressed
Novelty & positioning vs. prior work
Conceptual gap: simulating a given machine vs. learning task rules
Theoretical framing around CoT vs non-CoT
Evaluation/statistical robustness
Benchmark clarity and baselines

**Reviewer Scores:**

Likely only minor changes:
Reviewer m5Wv (4 → 4): concerns are conceptual; likely unchanged.
Reviewer wEq4 (6 → 6): concerns about statistics and baseline
Reviewer crJ8 (2 → 2): strong “does not engage with SOTA” stance
Reviewer b1Yx (4 → 4, or perhaps remains at 4): mostly robustness/writing/sample size

---

### Decision · Program_Chairs · 2026-01-26

Reject